# Design, Synthesis and Bioactive Evaluation of Topo I/*c-MYC* Dual Inhibitors to Inhibit Oral Cancer via Regulating the PI3K/AKT/NF-κB Signaling Pathway

**DOI:** 10.3390/molecules30040894

**Published:** 2025-02-14

**Authors:** Bin Zheng, Yi-Xiao Wang, Zi-Yan Wu, Xin-Wei Li, Li-Qing Qin, Nan-Ying Chen, Gui-Fa Su, Jun-Cheng Su, Cheng-Xue Pan

**Affiliations:** State Key Laboratory for Chemistry and Molecular Engineering of Medicinal Resources, Key Laboratory for Chemistry and Molecular Engineering of Medicinal Resources (Ministry of Education of China), Collaborative Innovation Center for Guangxi Ethnic Medicine, School of Chemistry and Pharmaceutical Sciences, Guangxi Normal University, 15 Yu Cai Road, Guilin 541004, China; zhengbin0611@163.com (B.Z.); wangyixiao2004@163.com (Y.-X.W.); 18150786271@163.com (Z.-Y.W.); lxw275397883@163.com (X.-W.L.); liqing20180621@163.com (L.-Q.Q.); chen_nanying@163.com (N.-Y.C.); gfysglgx@163.com (G.-F.S.); su_juncheng@163.com (J.-C.S.)

**Keywords:** Topo I inhibitor, *c-MYC*, PI3K/AKT/NF-κB, oral cancer, antitumor

## Abstract

The significantly rising incidence of oral cancer worldwide urgently requires the identification of novel, effective molecular targets to inhibit the progression of malignancy. DNA topoisomerase I (Topo I) is a well-established target for cancer treatment, and many studies have shown that different cancer cell genes could be targeted more selectively with one type of Topo I inhibitor. In this report, a new scaffold pyridothieno[3,2-*c*]isoquinoline 11,11-dioxide was designed via the combination of the key fragment or bioisoster of Topo I inhibitor azaindenoisoquinolines and G-quadruplex binder quindoline. Thirty-two target derivatives were synthesized, among which compounds **7be,** with potent Topo I inhibition, exhibited effective antiproliferative activity against Cal27, one of the oral cancer cell lines highly expressing Topo I protein. Further studies indicated that **7be** could also inhibit the activation of PI3K/AKT/NF-κB pathway and downregulate the level of c-MYC, repress the colony formation and the migration of Cal27 cells and trigger apoptosis and autophagy. Molecular docking indicated that **7be** could interact with the complex of Topo I and DNA via a mode similar to the indenoisoquinolines. The results of the Cal27 xenograft model confirmed that **7be** exhibited promising anticancer efficacy in vivo, with tumor growth inhibition (TGI) of 64.7% at 20 mg/kg.

## 1. Introduction

Oral cancer, the most commonly occurring malignant tumor among the head and neck malignancies [1], has been a significant public health risk with over 300,000 people diagnosed worldwide and around 140,000 deaths each year [2]. Surgical treatment and chemotherapy were the most frequently used therapy methods for oral cancer in clinical practice [2,3,4], but platinum-based drugs remain the only choice in chemotherapy, which suffers from negatives such as high toxicity, serious side effects and acquiring resistance. Therefore, the identification of a novel effective molecular target to inhibit the development and progression of oral cancer needs urgent attention [5].

DNA topoisomerase I (Topo I) has been validated as a promising target for anticancer agents, and a great deal of structurally diverse Topo I inhibitors have been reported [6]. The camptothecin derivatives (CPTs) topotecan (TPT **1**), irinotecan (**2**) and belotecan (**3**) have been used widely in clinics (Figure 1), and more than six other CPTs, including gimatecan, lurtotecan, exatecan, namitecan, silatecan and diflomotecan, have entered clinical trials. Besides the CPTs, some indolo[2,3-*a*]-carbazole derivatives such as edotecarin and J-107088, as well as three indenoisoquinoline derivatives, namely LMP400, LMP744 and LMP776 (Figure 1), also have undergone clinical trials for treatment of different kinds of solid cancers [7,8,9,10]. Many studies have shown that different cancer cell genes could be targeted more selectively with one type of Topo I inhibitor; it could be expected that certain Topo I inhibitors will be more suitable for different spectra of cancer [11,12,13]. Therefore, the Topo I might serve as an effective molecular target to inhibit the progression of oral cancer.

To date, thousands of compounds with potent Topo I inhibitory activity have been reported in the literature, but only three different scaffolds, namely the CPTs, indolo[2,3-*a*]-carbazoles and the indenoisoquinoline derivatives mentioned above, have successfully entered clinical application or trials at this stage [6,9]. Further analysis of the bioactive features of these scaffolds indicates that their derivatives could inhibit cancer cells via multiple mechanisms. For example, besides Topo I inhibition, most of the CPTs can also regulate the inflammation-related cell signaling pathway [14], while the indolocarbazoles can also serve as inhibitors of many cancer-related protein kinases [15], and the indenoisoquinoline derivatives exhibit potent activity in downregulating the *c-MYC* oncogene [16,17], suggesting that Topo I inhibitory scaffolds with multiple mechanisms might be more desirable for development into drug candidates.

Many azaindenoisoquinolines reported by Cushman’s group have been identified as promising anticancer molecules with multiple mechanisms. They can act not only as potent Topo I inhibitors, but some of them also possess inhibitory activities on tyrosyl-DNA phosphodiesterase (TDP) or binding to the *c-MYC* promoter G-Quadruplex and downregulating the transcription of this oncogene [12,18,19,20,21]. At the same time, many quindoline derivatives were found to be able to bind to and stabilize the G-quadruplex DNA [22,23,24,25,26,27], as well as to inhibit Topo I activity [28]. As there is continuous research interest in the Topo I inhibitors [29,30,31], herein we report the design, synthesis and bioactive evaluation of Topo I/*c-MYC* dual inhibitors based on the scaffolds of azaindenoisoquinoline and quindoline. These dual inhibitors could significantly suppress oral cancers via regulating the PI3K/AKT/NF-κB signaling pathway.

The design of the target compound is depicted in Figure 2. First, the polyaromatic ring skeleton of the target compound was created via fragment or bioisoster recombination. Next, a side chain, which has been proven to be very important for the Topo I activity of indenoisoquinoline and the G-quadruplex binding of quindoline, was linked to the skeleton at the position where it not only could enhance the G-quadruplex binding, but the Topo I inhibition as well. Finally, the carbonyl group on the azaindene ring fragment was replaced by a sulfonyl. Unlike the carbonyl group, the two oxygens of the sulfone protrude above the polyaromatic ring skeleton, which might help to decrease the planarity of the polyaromatic ring skeleton, since a fully planar polyaromatic unit is apt to intercalate into DNA resulting in unexpected side effects.

## 2. Results and Discussion

### 2.1. Chemistry

The synthesis of the target compounds **7** was outlined in Figure 1. To synthesize the 2-mercaptopyridine derivative **2**, sodium sulfide, sodium hydrosulfide and thiourea were used as sulfur sources to react with **1**, and thiourea was found to give the best results. Then **2** was reacted with methyl 2-(bromomethy)benzoate according to a published method [32] with slight optimization to give the intermediate **3**. Oxidation of **3** with hydrogen peroxide or *m*-chloroperoxybenzoic acid in different solvents was investigated. We found that the *m*-chloroperoxybenzoic acid in 1, 2-dichloroethane can smoothly oxidate **3** to **4**, but attempts to stop the reaction product at the sulfoxide were not successful. Compounds **4** were further converted to **5** via a tandem reaction that promoted potassium t-butoxide. Then, compounds **5** were reacted with POCl_3_ in toluene promoted by a catalytic quantity of DMF to provide the intermediate compounds **6**, which were further reacted with different amine derivatives to furnish the target compounds **7**. In this transformation step, the yield of most of the products was very low, which might be due to the high sensitivity to water of intermediates **6**, so any trace of water in the amine material could greatly affect the reaction. The structures of compounds **7** were characterized and confirmed by NMR and HRMS. As shown in Appendix A, **7ag**, **7ai** and **7be** were also determined by X-ray diffraction (CCDC: 2341050, 2405703 and 2341051; more data can be seen in Appendix A). 

### 2.2. Topo I Inhibition Assay

Firstly, an agarose gel electrophoresis assay was used to evaluate the Topo I inhibitory activity of the target compound **7**, and the results are outlined in Figure 3a. In the 20 μM concentration, only most of the **7bx** series of the target compounds exhibited obvious Topo I inhibition activity, among them compounds **7ba**, **7bc**, **7be**, **7bf** and **7bh** displaying potent Topo I activity while **7bb**, **7bd**, **7bg**, **7bf** and **7bi**, as well as most of the **7ax** and **7cx** series compounds, exhibited weak Topo I activity. In addition, in the lane of compounds **7ac**, **7af**, **7ag**, **7ah**, **7ak** and **7cf**, **7cg**, **7ch**, the nicked DNA could be clearly observed, suggesting that their presence can result in DNA breakage. The results of the agarose gel electrophoresis assay indicate that the substituent of R^1^, R^2^ and the side chain of R^3^ all can play an important role in Topo I activity. R^1^ = Me (**7bx** series) seems more favorable than R^1^ = H (**7ax** series) or = Ph Me (**7cx** series). The ω-substituents of the side chain R^3^ also can significantly affect the Topo I activity. In general, the dimethyl (**7ba**, **7bf**) or pyrrolidinyl (**7bc**, **7bh**) and piperidinyl (**7be**) can provide potent Topo I activity, while the morpholinyl (**7xd**, **7xi** series) and imidazole (**7xj** series) give very weak Topo I inhibition [33,34].

**7be** and **7bh**, the two compounds displaying the most potent activity, were further selected for testing of their Topo I potency at different concentrations. As seen in Figure 3b, with **7be** at 20 μM and **7bh** at 40 μM, the activity of Topo I would have been fully inhibited, making these more potent than the positive control compound CPT.

### 2.3. Cytotoxic Activity

Having evaluated their Topo I inhibitory activity, the antiproliferative activity of the target compounds against some common human cancer cell lines (NCI-H460, A549, ACHN, Cal27 and MGC-803) and one normal cell line (SV-HUC-1) was tested via an MTT assay. TPT, the first Topo I inhibitor approved in clinics, and cisplatin (CDDP), a commonly selected anticancer drug for treating head and neck malignancies, were used as the positive controls. As shown in Table 1, the compounds with potent Topo I inhibition, such as **7be**, **7bh** and **7cf,** also exhibited strong antiproliferative activity against the tested cell lines, suggesting that their cytotoxic activity might closely relate to their Topo I inhibitory activity. The other compounds exhibited no obvious Topo I activity, with **7ac**, **7af**, **7ah**, **7ak**, **7cf** and **7cg** also displaying moderate cytotoxicity, which might result from their capacity to damage the DNA. In the tested tumor cell lines, compound **7be** showed particularly significant antiproliferative activity against Cal27 cells, with an IC_50_ value as low as 1.12 μM, which is much stronger than the positive controls CDDP and TPT.

### 2.4. Structure–Activity Relationship (SAR) Study

After obtaining the activity of the compounds at the cellular level, preliminary SARs were carried out as summarized in Figure 4. We found that the antitumor activity of the compounds was relatively weak when R^1^ and R^2^ were H, and could significantly improve when methyl was introduced at the R^1^ and R^2^ positions. However, when a phenyl group was introduced into the R^2^ position, the activity and solubility (in DMSO) of the compounds would decrease dramatically. The side chain of R^3^ may be the most important substituent that can affect the activity of the target compound. Compounds with a piperidinyl or pyrrolidinyl at the end of the chain generally displayed potent antitumor and Topo I inhibitory activity, while those with a morpholinyl and imidazolyl possessed little cytotoxicity or Topo I inhibition. The length of the side chain could also affect the activity of the compounds. A chain with three methylene seemed more favorable than that with two methylene for the activity.

### 2.5. **7be** Exhibited Potent Cytotoxicity Against Cancer Cells with High Topo I Expression Levels

To determine the relationship between cytotoxicity and Topo I inhibition, the protein expression levels of Topo I in the tested cancer cell lines NCI-H460, A549, ACHN, Cal27 and MGC-803 and the normal SV-HUC-1 cell line were detected by protein blotting [35]. As seen in Figure 5, Cal27 and MGC-803, the two cell lines with the highest expression level of Topo I among the tested cells, are the cells against which the compound **7be** exhibited the strongest antiproliferation activity, with the IC_50_ value as low as 1.12 ± 0.72 μM and 1.69 ± 0.24 μM, respectively. Meanwhile, **7be** exhibited very weak cytotoxicity against the normal SV-HUC-1 cell line which expresses the lowest level of Topo I, suggesting that the antiproliferative activity of the compound **7be** is closely related to its Topo I inhibition.

### 2.6. Molecular Docking of **7be** with Topo I/DNA

To investigate the mechanism by which **7be** inhibits the Topo I, molecular docking was used to predict the binding modes of **7be** with the Topo I. As shown in Figure 6, compound **7be** could interact with Topo I via four hydrogen bonds. Among them, the sulfonyl group bound to the ARG-364 residue via two hydrogen bonds with distances of 2.2 and 2.5 Å, respectively. The other two hydrogen bonds were formed between the side chain of compound **7be** and the residue ASP-533 in distances of 2.0 and 3.4 Å. Meanwhile, the polyaromatic ring skeleton of **7be** also was able to be embedded into the double strands of the DNA. These results theoretically explain the ability of compound **7be** to bind to Topo I, indicating the important role of the sulfonyl group in compound **7be** for Topo I inhibition [36,37]. To verify the docking results, the indenoisoquinoline, a Topo I inhibitor scaffold reported by Cushman’s group with structural similarity to **7be,** also was docking with the same target, and these results indicated that the indenoisoquinoline did in fact give the same docking result reported in the literature [36].

### 2.7. Cellular Thermal Shift Assay (CETSA)

To further validate that compound **7be** could bind to Topo I, a cellular thermal shift assay (CETSA) [38,39], which is based on ligand-induced changes in the thermal stability of proteins, was carried out to assess the interactions between the drug and the target proteins in the cellular environment [40,41]. Oral cancer Cal27 cells were incubated with compound **7be** and then thermally denatured at different temperatures. The soluble fraction of the cell lysate was then collected and evaluated after freeze-thaw cycling. Binding of the Topo I protein to compound **7be** increases its overall resistance so that it can remain stable in the soluble lysate fraction at higher temperatures relative to the unbound protein.

As shown in Figure 7a,c, the thermal stability of the Topo I protein in the cell lysate was increased after the Cal27 cells were treated with 8 μM **7be** for 2 h, compared to the thermal stability of those from the Cal27 cells without **7be** intervention, especially at 52 to 56 °C. The CETSA analysis demonstrated that compound **7be** directly binds to and thermally stabilizes the Topo I protein outside the cell. At the same time, as shown in Figure 7b,d, when Cal27 cells were treated with 8 μM compound **7be** for 3 h, the increase in thermal stability of the Topo I protein in intact cells also could be observed, suggesting that compound **7be** could bind to and thermally stabilizes the Topo I protein directly within the cell.

### 2.8. **7be** Down-Regulated the Transcription of c-MYC and Its Expression Level

The target compounds in this research were designed as the combination of the key fragments of azaindenoisoquinolines and quindoline, two scaffolds with Topo I inhibitors and *c-MYC* promoter G-Quadruplex binders, to develop new Topo I scaffolds with multiple mechanisms. Having determined its Topo I inhibitory activity, we set out to investigate whether **7be** can also interfere with *c-MYC* to repress oncogenes [42]. RT-qPCR assays were first performed to determine the effects of **7be** on *c-MYC* transcription in Cal27 cells. As shown in Figure 8a, **7be** decreased the transcription of the *c-MYC* oncogene in a dose-dependent manner. Western blot assays were then conducted to evaluate the expression level of the c-MYC protein in Cal27 treated with **7be**. As shown in Figure 8b,c, a significant decline in the expression level of c-MYC protein could be observed in Cal27 cells after incubation with **7be**, indicating the rationality of our compound design and that **7be** could serve as a Topo I/*c-MYC* dual inhibitor.

Compared with the transcription and expression level of the *c-MYC* in Cal27 cells after treatment with compound **7be**, it could be found that it needed up to 8 μM **7be** to significantly inhibit the transcription of *c-MYC*, but needed only 2 μM of **7be** to significantly downregulate the level of c-MYC protein, suggesting that **7be** might not only be able to repress the transcription of *c-MYC* to decrease its expression but also use another mechanism to regulate the level of c-MYC.

### 2.9. **7be** Inhibited the PI3K/AKT/NF-κB Signaling Pathway

The PI3K/AKT pathway has an important impact on the occurrence and development of various cancers. Modulation of this signal pathway by some small molecules to regulate autophagy/apoptosis to suppress oral cancers has been observed [5,43,44]. PI3K/AKT also could regulate c-MYC and NF-κB which have been recognized as important factors closely related to the initiation, promotion and metastasis of tumors [45,46,47]. As the above results indicated that **7be** might possess another mechanism to regulate the level of c-MYC, in order to further explore the anticancer mechanism of compound **7be** on Cal27 cells, we further investigated the regulatory effects of **7be** on PI3K/AKT, the key upstream signal pathway of c-MYC. Western blotting was used to detect the protein expression and phosphorylation levels of PI3K and AKT, as well as the downstream NF-κB.

As shown in Figure 9, when Cal27 cells were treated with **7be**, their expression levels of PI3K were slightly increased, but the phosphorylation levels of PI3K and AKT were significantly repressed, which further resulted in the decrease of the expression and phosphorylation levels of NF-κB. These results suggested that **7be** can strongly inhibit the activation of the PI3K/AKT/NF-κB pathways in Cal27 cells, which might be the main mechanism of action for **7be** to exert its antitumor activity. In addition, the significant decrease in the expression level of the c-MYC in Cal27 cells at a low concentration of **7be** might mainly result from the inhibition of PI3K/AKT, a key protein upstream of c-MYC, but not the repression of transcription of *c-MYC*.

### 2.10. **7be** Inhibited the Colony Formation and Migration of Cal27 Cells

Cancer is one of the deadliest diseases due to its highly migratory and metastatic characteristics [48]. Compounds with a potent ability to inhibit the migration and metastasis of cancer cells are desirable for their long-term anticancer activity. To investigate the ability of compound **7be** to inhibit the metastasis and migration of cancer cells, colony formation experiments and wound healing tests were carried out. In the colony formation assay, Cal27 cells were treated with **7be** at concentrations of 0.5 μM, 1 μM and 1.5 μM, and in the wound healing tests treated with **7be** at 1 μM, 2 μM, 3 μM and 4 μM. As shown in Figure 10a,b, with the increase in **7be**, the number of cells decreased. The colony formation ability was completely suppressed by 1.5 μM **7be**, indicating that **7be** can inhibit clone formation. Furthermore, in Figure 10c,d, it could be seen that the gap in the control group is obviously narrower than that in the treated group, and with the increase in **7be**, the scraped space became wider and wider, indicating that **7be** could effectively inhibit migration and metastasis of Cal27 cells.

### 2.11. **7be** Arrested Cell Cycle at S Phase

Many anticancer compounds inhibit the proliferation of tumor cells by inhibiting the progression of the cell cycle [49]. In order to study how **7be** affects the cell cycle, flow cytometry was used to detect the cell cycle stage in cells treated with **7be**. As shown in Figure 11, when the dose of compound **7be** increased from 0 to 4 μM, the percentage of cells in the S phase gradually increased from 37.97% to 54.06%, indicating that **7be** could arrest the cell cycle at the S phase.

### 2.12. **7be** Induced Apoptosis

In the early stage of apoptosis, phosphatidylserine will turn from the inner cell membrane to the outside of the cell membrane, and Annexin-V can be detected by flow cytometry. Propidium Iodide (PI) is a kind of nuclear dye which cannot penetrate the complete cell membrane of living cells, but can dye the nucleus red through the cell membrane during late apoptosis and in dead cells.

In order to further study how compound **7be** induced cell death, we used the Annexin V-FITC/PI double-staining method to detect the apoptosis induced by **7be**. After Cal27 cells were treated with **7be** (1, 2, 3 and 4 μM), the number of apoptotic cells increased significantly after the intervention of **7be** at the concentration of 2 μM, and the number of apoptotic cells (early and late apoptotic cells) increased from 5.88% to 32.8% (Figure 12a). Based on the analysis results, **7be** showed obvious apoptosis in Cal27 cells, and the apoptosis rate was dose-dependent, indicating that **7be** could induce apoptosis in Cal27 cells.

At the same time, we used Hoechst 33258 staining to study whether **7be** can induce apoptosis. The cells in the control group were evenly stained and the weak intensity fluorescence was well distributed (Figure 12b). After Cal27 cells were treated with **7be** at different concentrations, obvious morphological changes occurred, such as cell contraction, membrane vesicle and chromatin condensation, nuclear destructive fragmentation, decrease in the number of adherent cells and so on, suggesting that **7be** induced apoptosis [50].

Resistance to apoptosis is a characteristic of tumor cells, and the activation of tumor cell apoptosis by small molecules provides a potentially useful way to improve the therapeutic response of tumors. **7be** up-regulated the pro-apoptotic protein Bax, down-regulated the anti-apoptotic protein Bcl-2 and up-regulated Cleaved PARP1/PARP1 (Figure 12c,d), further suggesting that **7be** may induce apoptosis of oral cancer Cal27 cells.

### 2.13. **7be** Induced Autophagy

Autophagy is a process in which cells are stimulated to swallow damaged organelles or cytoplasm and eventually degrade phagosomes in lysosomes [51,52]. Excessive autophagy is a pro-death mechanism of cancer cells. The process of autophagy requires the formation of a double-membrane vesicle called an autophagosome, which eventually fuses with the lysosome membrane to form an autophagy lysosome. Monodansylcadaverine (MDC) has been shown to be a marker of earlier autophagic compartments, and the accumulation of MDC-positive vesicles in cells corresponds to autophagy [53].

In order to determine whether compound **7be** induced autophagy, Cal27 cells were incubated with compound **7be** at different concentrations for 24 h. MDC staining was used to detect any MDC-positive vesicles in cells of Cal27 cells by confocal microscope (Figure 13a). Cal27 cells induced weak green fluorescence after being incubated with compound **7be** at low concentrations (1 μM and 2 μM) for 24 h. Unexpectedly, compound **7be** induced significant green fluorescence after incubation at high concentrations (3 μM and 4 μM) for 24 h, indicating that compound **7be** could trigger autophagy in Cal27 cells.

Meanwhile, to further confirm that compound **7be** could induce autophagic death in Cal27 cells, we used a classical autophagy inhibitor, chloroquine (CQ), to evaluate the role of autophagy in compound **7be**-mix-induced cell death. After Cal27 cells were treated with 20 μM CQ alone for 24 h, the cell survival rate was not significantly different from that of the control group, and the cell survival rate was also more than 80% after treatment with CQ alone for 48 h. When Cal27 cells were treated with 20 μM CQ and different concentrations of compound **7be**, the cell survival rate of the combined treatment group was higher than that of the group treated with compound **7be** alone. These data combined with the results of MDC staining suggest that compound **7be** can induce autophagic death of cells (Figure 13b,c).

In addition, in order to further clarify the potential mechanism of autophagy induced by compound **7be**, Western blot analysis was used to detect the expression of autophagy-related markers, including microtubule-associated protein 1 light chain protein-3 protein (LC3), p62 and Beclin-1. In the process of autophagy, LC3 is essential for the maturation of the autophagosome and its fusion with the lysosome. Under the mediation of members of the autophagy-associated protein (ATG) family, LC3 was cleaved to produce LC3-I, which further bound to phosphatidylethanolamine to form LC3-II and attached to the phagophore membrane until it fused with the phagophore membrane for 15 h to exhibit typical autophagic responses. The results showed that the expression of autophagy biomarkers Beclin-1 and LC3-II were upregulated, while the expression of p62 was decreased, further indicating that compound **7be** could induce autophagy in Cal27 cells (Figure 13d).

### 2.14. Cellular Localization Assay

Many anticancer compounds have been documented to localize in different subcellular organelles and cause cellular damage. To further confirm the intracellular distribution of compound **7be**, the localization of **7be** with the organelle-specific commercial stains Lyso-Tracker Deep Red (LTDR) and Mito-Tracker Deep Red FM (MTDR) in Cal27 cells was determined using confocal microscopy [54,55,56].

As shown in Figure 14a, the maximum absorption peak of **7be** in PBS was 350 nm, while the maximum emission peak of compound **7be** in PBS was 500 nm. Then, after Cal27 cells were treated with **7be** for 48 h, we used confocal microscopy at an excitation wavelength of 405 nm and received in the wavelength band of 415–515 nm, and **7be** could be observed to have crossed the cell membrane. As shown in Figure 14b, **7be** can effectively penetrate into Cal27 cells, and specifically localized in the lysosome. Pearson’s colocalization coefficient of **7be** with LTDR was determined to be 0.8. Meanwhile, a minimal colocalization coefficient (0.42) for **7be** and MTDR was observed. These results indicated that **7be** could possess lysosomal specificity. Therefore, **7be**-induced Cal27 cell death may be related to targeting lysosomes.

### 2.15. **7be** Displayed Effective Antitumor Activity In Vivo

In order to further investigate the therapeutic effects of compound **7be** in vivo, a model of a Cal27 xenograft tumor of human oral cancer was established, which was used to evaluate the antitumor activity of **7be** and the intensity of its effects on the human oral cancer Cal27 xenograft tumor model.

The results of the experiment showed that the relative tumor proliferation rate of the human oral cancer Cal27 xenograft tumor was 52.0%, and tumor suppression rate was 48.5% at 21 days of administration of the drug in the low-dose group at a dose of 10 mg/kg, which was administered intraperitoneally once every other day. When the high-dose group was administered the drug at a dose of 20 mg/kg by intraperitoneal injection once every other day for 21 days, the relative tumor proliferation rate was 34.1% and the tumor suppression rate was 64.7% against human oral cancer Cal27 xenograft tumors. In the cisplatin group, the drug dose was 2 mg/kg, injected intraperitoneally once every other day, and after 21 days of administration, the relative tumor proliferation rate of the Cal27 xenograft tumor of human oral cancer was 39.8%, and the tumor inhibition rate was 59.4%. Meanwhile, the experimental results showed that there was no significant decrease in body weight of BALB/c mice in all groups during the administration of the drug (*p* > 0.05). These data showed that **7be** had an obvious inhibitory effect on the growth of the Cal27 xenograft tumor of human oral cancer, and the order of antitumor activity against the Cal27 xenograft tumor of oral cancer was as follows: high dose group > cisplatin group > low dose group (Figure 15a,b,f).

Additionally, Western blot analysis of tumor tissue indicated that 20 mg/kg of **7be** decreased the expressions of p62, COX-2 and Bcl-2 levels in vivo (Figure 15c,d). These results suggest that **7be** suppresses oral tumor growth in vivo by modulating autophagy and promoting apoptosis.

To further evaluate the levels of proliferation and apoptotic death in the tumor tissues, hematoxylin and eosin (H&E) staining and Ki67 immunostaining assays were performed [57,58]. Analysis of the H&E-stained tumor sections suggested that more tumor cells in the **7be** group suffered a structural change with poor conditions compared with the other groups. The percentage of Ki67-positive cells decreased in the **7be**-treated group, indicating markedly reduced tumor cell proliferation after treatment (Figure 15e).

### 2.16. In Vivo Biological Safety Evaluation of Compound **7be**

Subsequently, to evaluate the side effects of compound **7be**, further studies were conducted by weighing and H&E staining the major organs [59,60]. The organ weight of mice showed that there was no significant difference in any of the groups (Figure 16a). The body weight of the mice was measured during treatment. As shown in Figure 16b, mice showed no considerable weight loss after any of the treatments. The H&E staining showed no lesions, degenerations or histopathological abnormalities (Figure 16c), indicating that **7be**-induced antitumor treatments were highly biocompatible. Collectively, these findings suggest that **7be** inhibits oral tumor growth with favorable tolerability in vivo.

## 3. Materials and Methods

### 3.1. General

All commercially available solvents and chemicals were used as received without further purification unless otherwise indicated. ^1^H and ^13^C- NMR spectra were recorded on Bruker AV-400 instruments (Bruker, Rheinstetten, Germany) with chemical shifts reported in ppm. Deuterated chloroform and deuterated dimethyl sulfoxide (DMSO-*d*) were used as the solvents and TMS as the internal standard. Melting points were recorded on an X-4B apparatus without correction. The high-resolution mass spectra (HRMS) were performed by an Agilent 6545 Q-TOF LC/MS (Agilent, Santa Clara, CA, USA), ^1^H and ^13^C NMR and HRMS spectra of compounds **2**–**7** are available in Appendix A.

### 3.2. Chemistry

#### 3.2.1. Synthesis of 2-Mercaptonicotinonitrile (**2a**)

Under electromagnetic stirring, compound **1a** (13.80 g, 20.0 mmol), thiourea (15.20 g, 40.0 mmol) and ethanol (60 mL) were added to a 250 mL round-bottom flask and heated in an 80 °C oil bath for 6 h (monitored by TLC, *V_EA_/V_PE_* = 2/1). The solvent was removed under reduced pressure, the thiourea was removed by washing with distilled water (3 × 20 mL), and 12.15 g yellow solid compound **2a** was obtained by drying in 89% yield.

##### 2-Mercaptonicotinonitrile (**2a**)

m.p. 202–203 °C. ^1^H NMR (400 MHz, DMSO-*d*_6_) *δ* 14.24 (s, 1H), 8.13 (dd, *J* = 7.4, 1.7 Hz, 1H), 7.95 (dd, *J* = 6.2, 1.8 Hz, 1H), 6.87 (dd, *J* = 7.4, 6.2 Hz, 1H). ^13^C NMR (100 MHz, DMSO-*d*_6_) *δ* 177.7, 146.0, 143.3, 117.3, 117.1, 112.9.

##### 2-Mercapto-4,6-dimethylnicotinonitrile (**2b**)

m.p. > 240 °C. ^1^H NMR (400 MHz, DMSO-*d*_6_) *δ* 13.85 (s, 1H), 6.69 (s, 1H), 2.34 (s, 3H), 2.33 (s, 3H). ^13^C NMR (100 MHz, DMSO-*d*_6_) *δ* 177.9, 157.4, 153.1, 116.8, 115.4, 113.9, 21.3, 19.3.

##### 2-Mercapto-4-methyl-6-phenylnicotinonitrile (**2c**)

m.p. 234–236 °C. ^1^H NMR (400 MHz, DMSO-*d*_6_) *δ* 14.02 (s, 1H), 7.81–7.74 (m, 2H), 7.62–7.51 (m, 3H), 7.13 (s, 1H), 2.45 (s, 3H). ^13^C NMR (100 MHz, DMSO-*d*_6_) *δ* 183.6, 179.1, 157.2, 152.1, 131.8, 129.2, 128.8, 116.8, 115.4, 114.6, 21.5.

#### 3.2.2. Synthesis of Methyl 2-(((3-Cyanopyridin-2-yl)thio)methyl)benzoate (**3a**)

Under electromagnetic stirring, methyl 2-bromo-methylbenzoate (2.28 g, 10.0 mmol), compound **2a** (1.36 g, 10.0 mmol), triethylamine (1.52 g, 15.0 mmol) and chloroform (20 mL) were added to a 100 mL round-bottom flask and heated to reflux for 3 h in an oil bath at 65 °C (monitored by TLC, *V_EA_/V_PE_* = 1/1). The solvent was removed under reduced pressure, washed with distilled water (3 × 20 mL) to remove triethylamine, filtrated and dried to give 2.46 g of a light-yellow solid compound **3a** in 86% yield.

##### Methyl 2-(((3-Cyanopyridin-2-yl)thio)methyl)benzoate (**3a**)

m.p. 114–116 °C. ^1^H NMR (400 MHz, DMSO-*d*_6_) *δ* 8.74 (dd, *J* = 5.0, 1.8 Hz, 1H), 8.22 (dd, *J* = 7.8, 1.8 Hz, 1H), 7.88 (dd, *J* = 7.8, 1.4 Hz, 1H), 7.64 (dd, *J* = 7.6, 1.4 Hz, 1H), 7.57–7.51 (m, 1H), 7.45–7.37 (m, 1H), 7.33 (dd, *J* = 7.7, 4.9 Hz, 1H). ^13^C NMR (100 MHz, DMSO-*d*_6_) *δ* 167.5, 161.5, 153.1, 142.4, 139.4, 133.0, 131.9, 130.9, 129.7, 128.3, 120.2, 115.9, 106.5, 52.7, 32.0. HRMS (ESI) *m*/*z*: calcd for C_15_H_13_N_2_O_2_S [M+H]^+^ 285.0692, found 285.0694.

##### Methyl 2-(((3-Cyano-4,6-dimethylpyridin-2-yl)thio)methyl)benzoate (**3b**)

m.p. 117–119 °C. ^1^H NMR (600 MHz, CDCl_3_) *δ* 7.96 (dd, *J* = 7.8, 1.5 Hz, 1H), 7.63 (dd, *J* = 7.7, 1.3 Hz, 1H), 7.43–7.39 (m, 1H), 7.32–7.27 (m, 1H), 6.77 (s, 1H), 4.93 (s, 2H), 3.94 (s, 3H), 2.56 (s, 3H), 2.43–2.35 (m, 3H). ^13^C NMR (150 MHz, CDCl_3_) *δ* 167.5, 162.1, 160.9, 152.2, 140.1, 132.0, 131.6, 131.1, 129.3, 127.4, 119.9, 115.0, 104.8, 52.2, 32.8, 24.6, 20.2. HRMS (ESI) *m*/*z*: calcd for C_17_H_17_N_2_O_2_S [M+H]^+^ 313.1010, found 313.1004.

##### Methyl 2-(((3-Cyano-4-methyl-6-phenylpyridin-2-yl)thio)methyl)benzoate (**3c**)

m.p. 115–117 °C. ^1^H NMR (400 MHz, DMSO-*d*_6_) *δ* 8.24–8.18 (m, 2H), 7.95–7.86 (m, 2H), 7.65–7.54 (m, 4H), 7.49 (d, *J* = 7.6Hz, 1H), 7.44–7.36 (m, 1H), 5.01 (s, 2H), 3.86 (s, 3H), 2.50 (s, 3H). ^13^C NMR (100 MHz, DMSO-*d*_6_) *δ* 167.4, 161.4, 158.0, 154.2, 139.5, 137.2, 133.0, 131.4, 131.2, 131.2, 129.6, 129.4, 128.3, 127.7, 117.8, 115.5, 105.4, 52.8, 32.5, 20.5. HRMS (ESI) *m*/*z*: calcd for C_22_H_19_N_2_O_2_S [M+H]^+^ 375.1162, found 375.1156.

#### 3.2.3. Synthesis of Methyl 2-(((3-Cyanopyridin-2-yl)sulfonyl)methyl)benzoate (**4a**)

Under electromagnetic stirring, compound **3a** (1.14 g, 4.0 mmol), 75% *m*-chloroperoxybenzoic acid (1.89 g, 8.2 mmol) and DCE (15 mL) were added sequentially to a 100 mL round-bottom flask and heated in an oil bath at 30 °C for 1 h (monitored by TLC, *V_EA_/V_PE_* = 1/1) and filtered. The residual *m*-chloroperoxybenzoic acid in the filtrate was removed with saturated Na_2_SO_3_ solution (3 × 10 mL) and the aqueous phase was extracted with DCM (3 × 20 mL). The organic phases were combined, and then washed with saturated sodium chloride solution, dried over anhydrous sodium sulfate and filtered. The solvent was removed under reduced pressure and purified by silica gel column chromatography (eluent, DCM) to give 0.77 g of a white solid compound **4a** in 61% yield.

#### 3.2.4. Synthesis of Pyrido[3′,2′:4,5]thieno[3,2-*c*]isoquinolin-5(6*H*)-one 11,11-dioxide (**5a**)

Under electromagnetic stirring, compound **4a** (0.77 g, 2.4 mmol), potassium tert-butanol (0.30 g, 2.7 mmol) and acetonitrile (15 mL) were added sequentially to a 100 mL round-bottom flask and heated in an oil bath at 40 °C for 1 h (monitored by TLC, *V_EA_/V_PE_* = 1/1). The solvent was removed under reduced pressure, 20 mL of distilled water was added to remove the residual potassium tert-butoxide, filtration was carried out, and the solid was purified by column chromatography on silica gel (eluent, *V_EA_/V_PE_* = 1/10) to give 0.31 g of compound **5a** as a light yellow solid in 45% yield.

##### Pyrido[3′,2′:4,5]thieno[3,2-*c*]isoquinolin-5(6*H*)-one 11,11-dioxide (**5a**)

m.p. > 300 °C. ^1^H NMR (400 MHz, DMSO-*d*_6_) *δ* 8.66 (dd, *J* = 4.9, 1.5 Hz, 1H), 8.30 (dd, *J* = 7.6, 1.5 Hz, 1H), 8.19–8.12 (m, 1H), 7.83–7.76 (m, 1H), 7.71 (dd, *J* = 7.7, 4.9 Hz, 1H), 7.66–7.56 (m, 2H). ^13^C NMR (100 MHz, DMSO-*d*_6_) *δ* 172.4, 159.0, 150.7, 131.7, 131.2, 130.4, 130.0, 129.4, 129.2, 128.7, 128.1, 125.1, 124.9, 120.8. HRMS (ESI) *m*/*z*: calcd for C_14_H_9_N_2_O_3_S [M+H]^+^ 285.0328, found 285.0329.

##### 7,9-Dimethylpyrido[3′,2′:4,5]thieno[3,2-*c*]isoquinolin-5(6*H*)-one 11,11-dioxide (**5b**)

m.p. > 330 °C. ^1^H NMR (400 MHz, DMSO-*d*_6_) *δ* 8.38–8.31 (m, 1H), 8.02–7.89 (m, 2H), 7.78–7.66 (m, 1H), 7.52 (s, 1H), 2.87 (s, 3H), 2.60 (s, 3H). ^13^C NMR (100 MHz, DMSO-*d*_6_) *δ* 161.3, 157.7, 156.6, 154.4, 148.5, 146.2, 138.2, 136.5, 134.9, 130.3, 129.0, 128.4, 122.3, 121.9, 23.8, 20.2. HRMS (ESI) *m*/*z*: calcd for C_16_H_13_N_2_O_3_S [M+H]^+^ 313.0641, found 313.0646.

##### 7-Methyl-9-phenylpyrido[3′,2′:4,5]thieno[3,2-*c*]isoquinolin-5(6*H*)-one 11,11-dioxide (**5c**)

m.p. > 300 °C. ^1^H NMR (600 MHz, DMSO-*d*_6_) *δ* 8.30 (dd, *J* = 7.9, 1.1 Hz, 1H), 8.23–8.19 (m, 3H), 7.90–7.82 (m, 2H), 7.64–7.49 (m, 5H), 2.99 (s, 3H). ^13^C NMR (150 MHz, DMSO-*d*_6_) *δ* 158.1, 157.3, 147.2, 136.8, 133.6, 131.0, 130.9, 130.4, 129.7, 129.6, 129.4, 128.5, 128.1, 127.7, 127.4, 126.8, 121.8, 115.1, 20.4. HRMS (ESI) *m*/*z*: calcd for C_21_H_15_N_2_O_3_S [M+H]^+^ 375.0803, found 375.0802.

#### 3.2.5. Synthesis of 5-Chloropyrido[3′,2′:4,5]thieno[3,2-*c*]isoquinoline 11,11-dioxide (**6a**)

Under electromagnetic stirring, compound **5a** (0.14 g, 0.5 mmol), POCl_3_ (1.53 g, 5 mmol), DMF (5 drops) and toluene (5 mL) were added sequentially to a 50 mL round-bottom flask, and the reaction was heated for 1 h in an oil bath at 110 °C (monitored by TLC, *V_EA_/V_PE_* = 1/1). Most of the toluene and phosphorous trichloride were removed under reduced pressure and purified by silica gel column chromatography to give 93.0 mg of compound **6a** as a white solid in 62% yield.

#### 3.2.6. General Procedure for Target Compounds **7aa**–**7al**, **7ba**–**7bj**, **7ca**–**7cj**

Under electromagnetic stirring, compound **6a** (75.5 mg, 0.25 mmol), N, N-dimethylethylenediamine (1.53 g, 5 mmol) and toluene (5 mL) were added sequentially to a 50 mL round-bottom flask and heated in an oil bath at 110 °C for 0.2 h (monitored by TLC, *V_EA_/V_PE_* = 1/1). Most of the toluene was removed under pressure and purified by silica gel column chromatography (eluent, first *V_EA_/V_PE_* = 1/1, then *V_DCM_/V_EA_/V_MeOH_* = 9/3/1). 65.4 mg of a light-yellow solid compound **7aa** was obtained with a yield of 73%.

##### 5-((2-(Dimethylamino)ethyl)amino)pyrido[3′,2′:4,5] thieno[3,2-*c*]isoquinoline 11,11-dioxide (**7aa**)

Yellow solid, 73% yield, m.p. 249–251 °C. ^1^H NMR (400 MHz, CDCl_3_) *δ* 8.68 (dd, *J* = 4.9, 1.6 Hz, 1H), 8.31 (dd, *J* = 7.7, 1.6 Hz, 1H), 8.22–8.15 (m, 1H), 7.86 (d, *J* = 8.4 Hz, 1H), 7.79–7.30 (m, 1H), 7.58–7.50 (m, 2H), 3.85–3.71 (m, 2H), 2.72–2.63 (m, 2H), 2.35 (s, 6H). ^13^C NMR (100 MHz, CDCl_3_) *δ* 158.8, 158.3, 150.9, 144.1, 132.4, 130.0, 129.9, 128.4, 127.6, 127.3, 123.5, 122.9, 117.8, 114.3, 57.2, 45.2, 38.9. HRMS (ESI) *m*/*z*: calcd for C_18_H_19_N_4_O_2_S [M+H]^+^ 355.1230, found 355.1219.

##### 5-((2-(Diethylamino)ethyl)amino)pyrido[3′,2′:4,5] thieno[3,2-*c*]isoquinoline 11,11-dioxide (**7ab**)

Yellow solid, 31% yield, m.p. 201–203 °C. ^1^H NMR (400 MHz, CDCl_3_) *δ* 8.68 (dd, *J* = 4.8, 1.5 Hz, 1H), 8.31 (dd, *J* = 7.7, 1.5 Hz, 1H), 8.19 (d, *J* = 8.2 Hz, 1H), 7.84–7.73 (m, 2H), 7.61–7.49 (m, 2H), 7.16–7.04 (m, 1H), 3.78–3.67 (m, 2H), 2.86–2.74 (m, 2H), 2.71–2.57 (m, 4H), 1.14–1.04 (m, 6H). ^13^C NMR (100 MHz, CDCl_3_) *δ* 158.8, 158.3, 150.9, 144.2, 132.3, 130.0, 128.5, 127.7, 127.2, 123.6, 122.7, 117.9, 114.2, 50.8, 46.7, 38.8, 12.1. HRMS (ESI) *m*/*z*: calcd for C_20_H_23_N_4_O_2_S [M+H]^+^ 383.1543, found 383.1541.

##### 5-((2-(Pyrrolidin-1-yl)ethyl)amino)pyrido[3′,2′:4,5] thieno[3,2-*c*]isoquinoline 11,11-dioxide (**7ac**)

Yellow solid, 10% yield, m.p. 283–285 °C. ^1^H NMR (400 MHz, DMSO-*d*_6_) *δ* 8.86–8.77 (m, 2H), 8.49–8.42 (m, 2H), 7.98–7.89 (m, 2H), 7.84 (dd, *J* = 7.8, 4.9 Hz, 1H), 7.75–7.68 (m, 1H), 3.94–3.81 (m, 2H), 2.90 (s, 2H), 2.70 (s, 4H), 1.80–1.67 (m, 4H). ^13^C NMR (100 MHz, DMSO-*d*_6_) *δ* 159.8, 158.0, 151.8, 144.3, 133.4, 130.9, 129.6, 129.0, 128.2, 127.9, 125.4, 122.4, 118.1, 112.7, 54.4, 54.2, 23.5. HRMS (ESI) *m*/*z*: calcd for C_20_H_21_N_4_O_2_S [M+H]^+^ 381.1387, found 381.1382.

##### 5-((2-Morpholinoethyl)amino)pyrido[3′,2′:4,5]thieno[3,2-*c*]isoquino-line 11,11-dioxide (**7ad**)

Yellow solid, 20% yield, m.p. 268–270 °C. ^1^H NMR (400 MHz, DMSO-*d*_6_) *δ* 8.78 (dd, *J* = 4.9, 1.4 Hz, 1H), 8.76–8.71 (m, 1H), 8.48–8.38 (m, 2H), 7.97–7.88 (m, 2H), 7.82 (dd, *J* = 7.7, 4.9 Hz, 1H), 7.74–7.67 (m, 1H), 3.87–3.78 (m, 2H), 3.61–3.51 (m, 4H), 2.71–2.61 (m, 2H), 2.54–2.46 (m, 4H).^13^C NMR (100 MHz, DMSO-*d*_6_) *δ* 159.8, 158.0, 151.8, 144.3, 133.4, 130.9, 129.6, 129.0, 128.2, 127.9, 125.3, 122.4, 118.0, 112.5, 66.7, 57.3, 53.9, 38.9. HRMS (ESI) *m*/*z*: calcd for C_20_H_21_N_4_O_3_S [M+H]^+^ 397.1336, found 397.1375.

##### 5-((2-(Piperidin-1-yl)ethyl)amino)pyrido[3′,2′:4,5] thieno[3,2-*c*]isoquinoline 11,11-dioxide (**7ae**)

Yellow solid, 15% yield, m.p. 235–237 °C. ^1^H NMR (400 MHz, CDCl_3_) *δ* 8.68 (dd, *J* = 4.9, 1.5 Hz, 1H), 8.31 (dd, *J* = 7.7, 1.6 Hz, 1H), 8.21 (d, *J* = 8.2 Hz, 1H), 7.88 (d, *J* = 8.4 Hz, 1H), 7.82–7.74 (m, 1H), 7.63–7.56 (m, 1H), 7.52 (dd, *J* = 7.7, 4.9 Hz, 1H), 3.88–3.76 (m, 2H), 2.83–2.67 (m, 2H), 2.57 (s, 4H), 1.74–1.63 (m, 4H), 1.53 (s, 2H). ^13^C NMR (100 MHz, CDCl_3_) *δ* 158.8, 158.3, 150.9, 144.1, 132.4, 130.0, 130.0, 128.4, 127.8, 127.2, 123.6, 122.8, 118.0, 114.4, 56.4, 54.3, 37.9, 25.9, 24.2. HRMS (ESI) *m*/*z*: calcd for C_21_H_23_N_4_O_2_S [M+H]^+^ 395.1543, found 395.1535.

##### 5-((3-(Dimethylamino)propyl)amino)pyrido[3′,2′:4,5] thieno[3,2-*c*]isoquinoline 11,11-dioxide (**7af**)

Yellow solid, 17% yield, m.p. 206–208 °C. ^1^H NMR (400 MHz, DMSO-*d*_6_) *δ* 8.97–8.89 (m, 1H), 8.78 (dd, *J* = 4.9, 1.5 Hz, 1H), 8.47–8.40 (m, 2H), 7.96–7.88 (m, 2H), 7.83 (dd, *J* = 7.7, 4.9 Hz, 1H), 7.73–7.67 (m, 1H), 3.79–3.68 (m, 2H), 2.57–2.47 (m, 2H), 2.28 (s, 6H), 1.96–1.85 (m, 2H). ^13^C NMR (100 MHz, DMSO-*d*_6_) *δ* 159.8, 158.0, 151.8, 144.3, 133.3, 130.9, 129.6, 128.9, 128.2, 127.9, 125.3, 122.4, 118.1, 112.5, 57.1, 45.1, 40.2, 26.2. HRMS (ESI) *m*/*z*: calcd for C_19_H_21_N_4_O_2_S [M+H]^+^ 369.1387, found 369.1377.

##### 5-((3-(Diethylamino)propyl)amino)pyrido[3′,2′:4,5] thieno[3,2-*c*]isoquinoline 11,11-dioxide (**7ag**)

Yellow solid, 13% yield, m.p. 194–196 °C. ^1^H NMR (400 MHz, DMSO-*d*_6_) *δ* 8.95–8.89 (m, 1H), 8.80 (dd, *J* = 4.9, 1.4 Hz, 1H), 8.46–8.38 (m, 2H), 7.98–7.88 (m, 2H), 7.84 (dd, *J* = 7.7, 4.9 Hz, 1H), 7.75–7.67 (m, 1H), 3.81–3.64 (m, 2H), 2.70–2.52 (m, 5H), 1.93–1.79 (m, 2H), 1.04–0.89 (m, 6H). ^13^C NMR (100 MHz, DMSO-*d*_6_) *δ* 159.7, 158.0, 151.7, 144.3, 133.3, 130.7, 129.6, 128.8, 128.1, 128.0, 125.2, 122.4, 118.1, 112.5, 50.6, 46.7, 40.5, 25.6, 11.5. HRMS (ESI) *m*/*z*: calcd for C_21_H_25_N_4_O_2_S [M+H]^+^ 397.1700, found 397.1691.

##### 5-((3-(Pyrrolidin-1-yl)propyl)amino)pyrido[3′,2′:4,5] thieno[3,2-*c*]isoquinoline 11,11-dioxide (**7ah**)

Yellow solid, 18% yield, m.p. 234–236 °C. ^1^H NMR (400 MHz, DMSO-*d*_6_) *δ* 9.08–9.01 (m, 1H), 8.78 (dd, *J* = 4.9, 1.5 Hz, 1H), 8.57–8.49 (m, 2H), 7.97–7.88 (m, 2H), 7.82 (dd, *J* = 7.7, 4.9 Hz, 1H), 7.74–7.67 (m, 1H), 3.83–3.71 (m, 2H), 3.31–3.07 (m, 6H), 2.21–2.04 (m, 2H), 1.94–1.85 (m, 4H). ^13^C NMR (100 MHz, DMSO-*d*_6_) *δ* 159.8, 158.0, 151.8, 144.3, 133.4, 131.1, 129.6, 128.9, 128.2, 127.9, 125.5, 122.4, 118.1, 112.8, 53.4, 52.5, 39.1, 25.6, 23.2. HRMS (ESI) *m*/*z*: calcd for C_21_H_23_N_4_O_2_S [M+H]^+^ 395.1543, found 395.1534.

##### 5-((3-Morpholinopropyl)amino)pyrido[3′,2′:4,5] thieno[3,2-*c*]isoquinoline 11,11-dioxide (**7ai**)

Yellow solid, 27% yield, m.p. 237–239 °C. ^1^H NMR (400 MHz, DMSO-*d*_6_) *δ* 8.83–8.75 (m, 2H), 8.44 (d, *J* = 8.5 Hz, 1H), 8.38 (dd, *J* = 7.7, 1.5 Hz, 1H), 7.97–7.86 (m, 2H), 7.83 (dd, *J* = 7.7, 4.9 Hz, 1H), 7.73–7.66 (m, 1H), 3.79–3.69 (m, 2H), 3.61–3.51 (m, 4H), 2.46–2.32 (m, 6H), 1.92–1.82 (m, 2H). ^13^C NMR (100 MHz, DMSO-*d*_6_) *δ* 159.8, 158.0, 151.8, 144.3, 133.3, 130.8, 129.6, 128.9, 128.1, 127.9, 125.3, 122.4, 118.0, 112.5, 66.7, 56.7, 53.9, 40.4, 25.6. HRMS (ESI) *m*/*z*: calcd for C_21_H_23_N_4_O_3_S [M+H]^+^ 411.1493, found 411.1489.

##### 5-((3-(1H-Imidazol-1-yl)propyl)amino)pyrido[3′,2′:4,5]thieno[3,2-*c*]isoquinoline 11,11-dioxide (**7aj**)

Yellow solid, 31% yield, m.p. 277–279 °C. ^1^H NMR (400 MHz, DMSO-*d*_6_) *δ* 8.86–8.74 (m, 2H), 8.45 (d, *J* = 8.4 Hz, 1H), 8.36 (dd, *J* = 7.7, 1.5 Hz, 1H), 7.97–7.87 (m, 2H), 7.83 (dd, *J* = 7.7, 4.9 Hz, 1H), 7.75–7.66 (m, 2H), 7.25 (d, *J* = 1.3 Hz, 1H), 6.94 (d, *J* = 1.2 Hz, 1H), 4.17–4.06 (m, 2H), 3.73–3.58 (m, 2H), 2.22–2.11 (m, 2H). ^13^C NMR (100 MHz, DMSO-*d*_6_) *δ* 159.8, 158.0, 151.8, 144.2, 137.9, 133.4, 131.0, 129.6, 128.9, 128.2, 127.9, 125.4, 122.4, 119.9, 118.1, 112.7, 44.4, 39.1, 30.5. HRMS (ESI) *m*/*z*: calcd for C_20_H_18_N_5_O_2_S [M+H]^+^ 392.1183, found 392.1175.

##### 5-((3-Aminopropyl)amino)pyrido[3′,2′:4,5]thieno[3,2-*c*]isoquinoline 11,11-dioxide (**7ak**)

Yellow solid, 60% yield, m.p. 199–201 °C. ^1^H NMR (400 MHz, DMSO-*d*_6_) *δ* 8.79 (dd, *J* = 4.9, 1.5 Hz, 1H), 8.59 (d, *J* = 8.4 Hz, 1H), 8.53 (dd, *J* = 7.7, 1.5 Hz, 1H), 7.97–7.88 (m, 2H), 7.84 (dd, *J* = 7.8, 4.9 Hz, 1H), 7.73–7.67 (m, 1H), 3.84–3.71 (m, 2H), 2.95–2.85 (m, 2H), 2.08–1.97 (m, 2H). ^13^C NMR (100 MHz, DMSO-*d*_6_) *δ* 159.8, 158.0, 151.8, 144.3, 133.4, 131.2, 129.6, 128.9, 128.2, 127.9, 125.6, 122.4, 118.1, 112.7, 38.8, 37.5, 27.4. HRMS (ESI) *m*/*z*: calcd for C_17_H_17_N_4_O_2_S [M+H]^+^ 341.1074, found 341.1066.

##### 5-(Pyrrolidin-1-yl)pyrido[3′,2′:4,5]thieno[3,2-*c*]isoquinoline 11,11-dioxide (**7al**)

Yellow solid, 5% yield, m.p. 289–291 °C. ^1^H NMR (400 MHz, DMSO-*d*_6_) *δ* 8.78 (dd, *J* = 5.0, 1.5 Hz, 1H), 8.51–8.38 (m, 2H), 7.97–7.86 (m, 2H), 7.81 (dd, *J* = 7.7, 4.9 Hz, 1H), 7.67–7.59 (m, 1H), 4.00 (d, *J* = 6.5 Hz, 4H), 1.99 (d, *J* = 3.5 Hz, 4H). ^13^C NMR (100 MHz, DMSO-*d*_6_) *δ* 160.1, 158.2, 151.8, 143.4, 132.9, 131.4, 130.9, 128.9, 128.7, 127.8, 126.9, 121.9, 118.7, 112.6, 52.2, 25.7. HRMS (ESI) *m*/*z*: calcd for C_18_H_16_N_3_O_2_S [M+H]^+^ 338.0965, found 338.0961.

##### 5-((2-(Dimethylamino)ethyl)amino)-7,9-dimethylpyrido[3′,2′:4,5]thieno[3,2-*c*]isoquinoline 11,11-dioxide (**7ba**)

Yellow solid, 10% yield, m.p. 273–275 °C. ^1^H NMR (400 MHz, DMSO-*d*_6_) *δ* 9.08–8.99 (m, 1H), 8.57 (d, *J* = 8.4 Hz, 1H), 7.97–7.86 (m, 2H), 7.72–7.63 (m, 1H), 7.47 (s, 1H), 3.99–3.86 (m, 2H), 3.18 (s, 2H), 2.89 (s, 3H), 2.63 (s, 6H), 2.58 (s, 3H). ^13^C NMR (100 MHz, DMSO-*d*_6_) *δ* 160.5, 159.3, 157.8, 146.6, 146.5, 133.3, 130.1, 129.7, 127.9, 125.6, 122.7, 122.2, 117.1, 112.1, 56.2, 43.9, 38.3, 23.9, 21.6, 20.1. HRMS (ESI) *m*/*z*: calcd for C_20_H_23_N_4_O_2_S [M+H]^+^ 383.1543, found 383.1537.

##### 5-((2-(Diethylamino)ethyl)amino)-7,9-dimethylpyrido[3′,2′:4,5]thieno[3,2-*c*]isoquinoline 11,11-dioxide (**7bb**)

Yellow solid, 15% yield, m.p. 195–197 °C. ^1^H NMR (400 MHz, CDCl_3_) *δ* 8.21–8.14 (m, 1H), 7.82–7.68 (m, 2H), 7.56–7.48 (m, 1H), 7.14 (s, 1H), 7.05 (d, *J* = 5.5 Hz, 1H), 3.72–3.62 (m, 2H), 2.90 (s, 3H), 2.83–2.74 (m, 2H), 2.69–2.58 (m, 7H), 1.12–1.03 (m, 6H). ^13^C NMR (100 MHz, CDCl_3_) *δ* 160.3, 158.3, 158.1, 146.9, 146.0, 132.2, 130.2, 129.4, 127.2, 123.4, 123.2, 122.5, 116.7, 113.1, 50.8, 46.7, 39.1, 24.0, 19.9, 12.1. HRMS (ESI) *m*/*z*: calcd for C_22_H_27_N_4_O_2_S [M+H]^+^ 411.1856, found 411.1851.

##### 7,9-Dimethyl-5-((2-(pyrrolidin-1-yl)ethyl)amino) pyrido[3′,2′:4,5]thieno[3,2-*c*]isoquinoline 11,11-dioxide (**7bc**)

Yellow solid, 16% yield, m.p. 255–257 °C. ^1^H NMR (400 MHz, CDCl_3_) *δ* 8.21–8.15 (m, 1H), 7.84 (d, *J* = 8.4 Hz, 1H), 7.76–7.69 (m, 1H), 7.55–7.47 (m, 1H), 7.14 (s, 1H), 6.96–6.86 (m, 1H), 3.84–3.66 (m, 2H), 2.94–2.86 (m, 5H), 2.70–2.60 (m, 7H), 1.91–1.82 (m, 4H). ^13^C NMR (100 MHz, CDCl_3_) *δ* 160.3, 158.4, 158.0, 146.8, 146.0, 132.2, 130.1, 129.4, 127.2, 123.4, 123.2, 122.7, 116.6, 113.2, 54.1, 54.0, 40.5, 24.0, 23.6, 19.9. HRMS (ESI) *m*/*z*: calcd for C_22_H_25_N_4_O_2_S [M+H]^+^ 409.1700, found 409.1692.

##### 7,9-Dimethyl-5-((2-morpholinoethyl)amino)pyrido[3′,2′:4,5]thieno[3,2-*c*]isoquinoline 11,11-dioxide (**7bd**)

Yellow solid, 14% yield, m.p. 298–299 °C. ^1^H NMR (400 MHz, DMSO-*d*_6_) *δ* 8.73–8.67 (m, 1H), 8.41 (d, *J* = 8.4 Hz, 1H), 7.95–7.85 (m, 2H), 7.71–7.61 (m, 1H), 7.47 (s, 1H), 3.81–3.71 (m, 2H), 3.62–3.52 (m, 4H), 2.91 (s, 3H), 2.71–2.62 (m, 2H), 2.58 (s, 3H), 2.47 (d, *J* = 4.7 Hz, 4H). ^13^C NMR (100 MHz, DMSO-*d*_6_) *δ* 160.4, 159.3, 157.9, 146.8, 146.6, 133.2, 130.1, 129.7, 125.1, 122.8, 122.3, 116.8, 66.7, 57.0, 53.9, 23.9, 19.9. HRMS (ESI) *m*/*z*: calcd for C_22_H_25_N_4_O_3_S [M+H]^+^ 425.1649, found 425.1644.

##### 7,9-Dimethyl-5-((2-(piperidin-1-yl)ethyl)amino) pyrido[3′,2′:4,5]thieno[3,2-*c*]isoquinoline 11,11-dioxide (**7be**)

Yellow solid, 14% yield, m.p. 265–267 °C. ^1^H NMR (400 MHz, DMSO-*d*_6_) *δ* 8.72–8.62 (m, 1H), 8.39 (d, *J* = 8.4 Hz, 1H), 7.95–7.84 (m, 2H), 7.71–7.61 (m, 1H), 7.44 (s, 1H), 3.79–3.65 (m, 2H), 2.89 (s, 3H), 2.67–2.53 (m, 5H), 2.41 (s, 4H), 1.54–1.43 (m, 4H), 1.41–1.32 (m, 2H). ^13^C NMR (100 MHz, DMSO-*d*_6_) *δ* 160.4, 159.3, 157.9, 146.8, 146.5, 133.2, 130.1, 129.7, 127.8, 125.1, 122.8, 122.2, 116.8, 111.6, 57.3, 54.7, 26.1, 24.5, 23.9, 19.8. HRMS (ESI) *m*/*z*: calcd for C_23_H_27_N_4_O_2_S [M+H]^+^ 423.1856, found 423.1854.

##### 5-((3-(Dimethylamino)propyl)amino)-7,9-dimethylpyrido[3′,2′:4,5]thieno[3,2-*c*]isoquinoline 11,11-dioxide (**7bf**)

Yellow solid, 12% yield, m.p. 193–195 °C. ^1^H NMR (400 MHz, CDCl_3_) *δ* 9.13 (s, 1H), 8.21–8.12 (m, 1H), 7.74–7.64 (m, 2H), 7.52–7.42 (m, 1H), 7.13 (s, 1H), 3.82–3.72 (m, 2H), 2.90 (s, 3H), 2.67–2.59 (m, 5H), 2.40 (s, 6H), 1.97–1.84 (m, 2H). ^13^C NMR (100 MHz, CDCl_3_) *δ* 160.1, 158.9, 158.2, 147.2, 146.0, 131.9, 130.2, 129.3, 127.0, 123.4, 123.3, 122.8, 117.0, 112.4, 59.8, 45.5, 43.5, 24.3, 24.0, 19.9. HRMS (ESI) *m*/*z*: calcd for C_21_H_25_N_4_O_2_S [M+H]^+^ 397.1700, found 397.1694.

##### 5-((3-(Diethylamino)propyl)amino)-7,9-dimethyl-pyrido[3′,2′:4,5]thieno[3,2-*c*]isoquinoline 11,11-dioxide (**7bg**)

Yellow solid, 14% yield, m.p. 195–197 °C. ^1^H NMR (400 MHz, CDCl_3_) *δ* 9.32 (s, 1H), 8.17 (dd, *J* = 8.3, 1.2 Hz, 1H), 7.80 (d, *J* = 8.4 Hz, 1H), 7.75–7.65 (m, 1H), 7.53–7.41 (m, 1H), 7.14 (s, 1H), 3.83–3.73 (m, 2H), 2.91 (s, 3H), 2.76–2.65 (m, 6H), 2.64 (s, 3H), 1.96–1.87 (m, 2H), 1.17–1.08 (m, 6H). ^13^C NMR (100 MHz, DMSO-*d*_6_) *δ* 160.4, 159.3, 157.9, 146.9, 146.5, 133.1, 130.1, 129.7, 127.7, 125.0, 122.8, 122.2, 116.8, 111.4, 50.8, 46.8, 41.1, 26.2, 23.9, 19.7, 12.1. HRMS (ESI) *m*/*z*: calcd for C_23_H_29_N_4_O_2_S [M+H]^+^ 425.2013, found 425.2007.

##### 7,9-Dimethyl-5-((3-(pyrrolidin-1-yl)propyl)amino)-pyrido[3′,2′:4,5]thieno[3,2-*c*]isoquinoline 11,11-dioxide (**7bh**)

Yellow solid, 16% yield, m.p. 241–243 °C. ^1^H NMR (400 MHz, CDCl_3_) *δ* 9.37 (d, *J* = 4.2 Hz, 1H), 8.21–8.12 (m, 1H), 7.75–7.64 (m, 2H), 7.50–7.40 (m, 1H), 7.13 (s, 1H), 3.82–3.73 (m, 2H), 2.90 (s, 3H), 2.86–2.79 (m, 2H), 2.71–2.65 (m, 4H), 2.64 (s, 3H), 2.00–1.90 (m, 6H). ^13^C NMR (100 MHz, CDCl_3_) *δ* 160.1, 159.0, 158.2, 147.3, 146.0, 131.9, 130.2, 129.3, 126.6, 123.4, 123.3, 122.9, 117.0, 112.3, 56.5, 54.3, 43.9, 25.4, 24.0, 23.6, 19.9. HRMS (ESI) *m*/*z*: calcd for C_23_H_27_N_4_O_2_S [M+H]^+^ 423.1856, found 423.1850.

##### 7,9-Dimethyl-5-((3-morpholinopropyl)amino)-pyrido[3′,2′:4,5]thieno[3,2-*c*]isoquinoline 11,11-dioxide (**7bi**)

Yellow solid, 10% yield, m.p. 222–224 °C. ^1^H NMR (400 MHz, CDCl_3_) *δ* 8.31–8.22 (m, 1H), 8.21–8.13 (m, 1H), 7.93 (d, *J* = 8.4 Hz, 1H), 7.78–7.68 (m, 1H), 7.54–7.44 (m, 1H), 7.15 (s, 1H), 3.88–3.82 (m, 4H), 3.81–3.75 (m, 2H), 2.90 (s, 3H), 2.70–2.60 (m, 2H), 2.65 (s, 3H), 2.60 (s, 4H), 2.00–1.90 (m, 2H). ^13^C NMR (100 MHz, CDCl_3_) *δ* 160.3, 158.7, 158.1, 147.0, 146.0, 132.1, 130.1, 129.4, 126.9, 123.5, 123.3, 122.9, 116.7, 112.8, 67.0, 59.3, 54.0, 24.0, 23.3, 19.9, 0.01. HRMS (ESI) *m*/*z*: calcd for C_23_H_27_N_4_O_3_S [M+H]^+^ 439.1806, found 439.1801.

##### 5-((3-(1H-Imidazol-1-yl)propyl)amino)-7,9-dimethylpyrido[3′,2′:4,5]thieno[3,2-*c*]isoquinoline 11,11-dioxide (**7bj**)

Yellow solid, 12% yield, m.p. > 300 °C. ^1^H NMR (400 MHz, DMSO-*d*_6_) *δ* 8.75–8.63 (m, 1H), 8.44 (d, *J* = 8.4 Hz, 1H), 7.96–7.85 (m, 2H), 7.73–7.64 (m, 2H), 7.46 (s, 1H), 7.27–7.19 (m, 1H), 6.91–6.87 (m, 1H), 4.14–3.98 (m, 2H), 3.67–3.51 (m, 2H), 2.80 (s, 3H), 2.58 (s, 3H), 2.22–2.10 (m, 2H). ^13^C NMR (100 MHz, DMSO-*d*_6_) *δ* 160.4, 159.4, 157.8, 146.7, 146.6, 137.8, 133.2, 130.2, 129.7, 128.9, 127.8, 125.2, 122.8, 122.2, 119.9, 116.9, 111.7, 44.4, 30.2, 23.9, 19.9. HRMS (ESI) *m*/*z*: calcd for C_22_H_22_N_5_O_2_S [M+H]^+^ 420.1496, found 420.1490.

##### 5-((2-(Dimethylamino)ethyl)amino)-7-methyl-9-phenylpyrido[3′,2′:4,5]-thieno[3,2-*c*]isoquinoline 11,11-dioxide (**7ca**)

Yellow solid, 14% yield, m.p. > 300 °C. HRMS (ESI) *m*/*z*: calcd for C_25_H_25_N_4_O_2_S [M+H]^+^ 445.1700, found 445.1694.

##### 5-((2-(Diethylamino)ethyl)amino)-7-methyl-9-phenylpyrido[3′,2′:4,5]thieno[3,2-*c*]isoquinoline 11,11-dioxide (**7cb**)

Yellow solid, 28% yield, m.p. > 300 °C. ^1^H NMR (600 MHz, DMSO-*d*_6_) *δ* 8.52 (d, *J* = 8.4 Hz, 1H), 8.25–8.16 (m, 3H), 7.98–7.87 (m, 2H), 7.72–7.64 (m, 1H), 7.59–7.50 (m, 3H), 3.89–3.76 (m, 2H), 3.02 (s, 3H), 2.52–2.45 (m, 6H), 1.04–0.98 (m, 6H). ^13^C NMR (150 MHz, DMSO-*d*_6_) *δ* 158.3, 157.4, 156.2, 146.5, 145.5, 135.8, 132.2, 129.7, 128.6, 128.5, 127.0, 126.3, 125.9, 124.3, 122.8, 121.2, 116.0, 111.1, 50.2, 46.1, 45.9, 20.8, 19.2. HRMS (ESI) *m*/*z*: calcd for C_27_H_29_N_4_O_2_S [M+H]^+^ 473.1933, found 473.2006.

##### 7-Methyl-9-phenyl-5-((2-(pyrrolidin-1-yl)ethyl)amino)pyrido[3′,2′:4,5]thieno[3,2-*c*]isoquinoline 11,11-dioxide (**7cc**)

Yellow solid, 13% yield, m.p. > 300 °C. HRMS (ESI) *m*/*z*: calcd for C_27_H_27_N_4_O_2_S [M+H]^+^ 471.1856, found 471.1856.

##### 7-Methyl-5-((2-morpholinoethyl)amino)-9-phenylpyrido[3′,2′:4,5]thieno[3,2-*c*]isoquinoline 11,11-dioxide (**7cd**)

Yellow solid, 15% yield, m.p. > 300 °C. HRMS (ESI) *m*/*z*: calcd for C_27_H_27_N_4_O_3_S [M+H]^+^ 487.1806, found 487.1799.

##### 7-Methyl-9-phenyl-5-((2-(piperidin-1-yl)ethyl)-amino)pyrido[3′,2′:4,5]thieno[3,2-*c*]isoquinoline 11,11-dioxide (**7ce**)

Yellow solid, 23% yield, m.p. 200–202 °C. HRMS (ESI) *m*/*z*: calcd for C_28_H_29_N_4_O_2_S [M+H]^+^ 485.2013, found 485.2008.

##### 5-((3-(Dimethylamino)propyl)amino)-7-methyl-9-phenylpyrido[3′,2′:4,5]thieno[3,2-*c*]isoquinoline 11,11-dioxide (**7cf**)

Yellow solid, 21% yield, m.p. 220–222 °C. ^1^H NMR (600 MHz, DMSO-*d*_6_) *δ* 8.92–8.81 (m, 1H), 8.41 (d, *J* = 8.5 Hz, 1H), 8.24–8.17 (m, 3H), 8.00–7.88 (m, 2H), 7.73–7.68 (m, 1H), 7.61–7.50 (m, 3H), 3.72–3.61 (m, 2H), 3.01 (s, 3H), 2.61–2.52 (m, 2H), 2.34 (s, 6H), 1.97–1.91 (m, 2H). ^13^C NMR (150 MHz, DMSO-*d*_6_) *δ* 159.3, 158.5, 157.3, 147.6, 146.6, 136.9, 133.3, 130.8, 129.7, 129.5, 128.0, 127.3, 126.9, 125.1, 123.9, 122.4, 117.0, 112.1, 56.9, 44.9, 25.9, 21.7, 20.2. HRMS (ESI) *m*/*z*: calcd for C_26_H_27_N_4_O_2_S [M+H]^+^ 459.1856, found 459.1847.

##### 5-((3-(Diethylamino)propyl)amino)-7-methyl-9-phenylpyrido[3′,2′:4,5]thieno[3,2-*c*]isoquinoline 11,11-dioxide (**7cg**)

Yellow solid, 19% yield, m.p. 244–246 °C. ^1^H NMR (400 MHz, DMSO-*d*_6_) *δ* 8.94–8.85 (m, 1H), 8.40 (d, *J* = 8.4 Hz, 1H), 8.26–8.17 (m, 3H), 7.99–7.87 (m, 2H), 7.75–7.66 (m, 1H), 7.62–7.50 (m, 3H), 3.78–3.61 (m, 2H), 3.04 (s, 3H), 2.57–2.51 (m, 2H), 2.49–2.45 (m, 4H), 1.94–1.78 (m, 2H), 1.02–0.85 (m, 6H). ^13^C NMR (150 MHz, DMSO-*d*_6_) *δ* 159.3, 158.6, 157.3, 147.5, 146.7, 136.9, 133.2, 130.8, 129.7, 129.6, 128.0, 127.4, 127.0, 125.0, 123.9, 122.4, 116.9, 112.0, 50.8, 46.8, 41.1, 26.3, 20.1, 12.2. HRMS (ESI) *m*/*z*: calcd for C_28_H_31_N_4_O_2_S [M+H]^+^ 487.2169, found 487.2168.

##### 7-Methyl-9-phenyl-5-((3-(pyrrolidin-1-yl)propyl)-amino)pyrido[3′,2′:4,5]thieno[3,2-*c*]isoquinoline 11,11-dioxide (**7ch**)

Yellow solid, 23% yield, m.p. 293–295 °C. ^1^H NMR (600 MHz, DMSO-*d*_6_) *δ* 8.93 (s, 1H), 8.38 (d, *J* = 8.4 Hz, 1H), 8.25–8.15 (m, 3H), 7.99–7.86 (m, 2H), 7.76–7.62 (m, 1H), 7.61–7.48 (m, 3H), 3.78–3.65 (m, 2H), 3.02 (s, 3H), 2.56–2.50 (m, 2H), 2.44 (s, 4H), 1.94–1.85 (m, 2H), 1.69 (s, 4H). ^13^C NMR (150 MHz, DMSO-*d*_6_) *δ* 159.4, 158.6, 157.3, 147.6, 146.7, 136.9, 133.3, 130.8, 129.7, 129.6, 128.0, 127.4, 127.0, 125.0, 123.9, 122.4, 117.0, 112.0, 54.2, 41.0, 40.5, 27.9, 23.6, 20.2. HRMS (ESI) *m*/*z*: calcd for C_28_H_29_N_4_O_2_S [M+H]^+^ 485.2013, found 485.2008.

##### 7-Methyl-5-((3-morpholinopropyl)amino)-9-phenylpyrido[3′,2′:4,5]thieno[3,2-*c*]isoquinoline 11,11-dioxide (**7ci**)

Yellow solid, 17% yield, m.p. > 300 °C. ^1^H NMR (400 MHz, DMSO-*d*_6_) *δ* 8.84–8.76 (m, 1H), 8.44 (d, *J* = 8.5 Hz, 1H), 8.26–8.17 (m, 3H), 8.00–7.86 (m, 2H), 7.74–7.65 (m, 1H), 7.63–7.49 (m, 3H), 3.78–3.66 (m, 2H), 3.59–3.49 (m, 4H), 3.04 (s, 3H), 2.46–2.30 (m, 6H), 1.95–1.86 (m, 2H). ^13^C NMR (100 MHz, DMSO-*d*_6_) *δ* 160.0, 159.3, 158.6, 157.3, 147.5, 146.7, 136.9, 133.3, 130.8, 129.7, 129.6, 127.4, 127.0, 125.1, 123.9, 122.4, 117.0, 112.0, 66.7, 56.8, 53.9, 25.7, 20.2. HRMS (ESI) *m*/*z*: calcd for C_28_H_29_N_4_O_3_S [M+H]^+^ 501.1962, found 501.1962.

##### 5-((3-(1H-imidazol-1-yl)propyl)amino)-7-methyl-9-phenylpyrido[3′,2′:4,5]thieno[3,2-*c*]isoquinoline 11,11-dioxide (**7cj**)

Yellow solid, 23% yield, m.p. > 300 °C. HRMS (ESI) *m*/*z*: calcd for C_27_H_24_N_5_O_2_S [M+H]^+^ 482.1652, found 482.1651.

### 3.3. Biological Assay Methods

#### 3.3.1. MTT Assay for Cell Viability

Human oral cancer Cal27 cells were seeded in 96-well plates and incubated overnight in 180 μL of medium containing 10% Fetal Bovine Serum (Gibco, San Francisco, CA, USA). On the following day, compounds at final concentrations of 2.5, 5, 10 and 20 μM were added to each well and incubated for 48 h. Then, 10 μL of 5 mg/mL MTT solution was added to each well and incubated at 37 °C for 4 h. After 4 h of incubation at 37 °C, the MTT-containing medium was removed and DMSO was added to each well and the absorbance at 570 nm and 630 nm was measured using a microplate reader (CYTATION 3, Burton, Westwood, NJ, USA). Survival/% = (mean OD of experimental group/mean OD of control group) × 100%. Each test was repeated at least three times.

#### 3.3.2. Molecular Docking

Molecular docking was performed in AutoDockTools-1.5.7 software. Crystal structures of Topo I protein were downloaded from the Protein Data Bank (PDB ID: 1T8I). Small molecules in the crystal structures were removed and polar hydrogens were added using AutoDockTool-1.5.7 to dock the compound molecules to the binding sites of the ligands in the PDB file. Docking runs and subsequent scoring were performed using the standard parameters of the program. Images were polished with PyMOL2.5.7.

#### 3.3.3. Western Blot Analysis

After Cal27 cells had been treated with the desired concentration of each drug for 24 h, the cells were collected and the total protein was extracted. The cells were then separated on a gel of appropriate concentration and transferred to a PVDF membrane. After blocking with milk, the membrane was incubated with primary antibody at 4 °C overnight. Subsequently, the membrane was incubated with horseradish peroxidase-coupled secondary antibody for 2 h. Finally, the protein bands were used to detect the immunoreactive signal using an enhanced chemiluminescence kit.

#### 3.3.4. Topo I Inhibitory Activity Assay

In this assay, 250 ng of pBR322 plasmid DNA, different concentrations of the compounds to be tested, 0.1% BSA 1 μL, 10× Topo I buffer, Topo I and add 1× TAE buffer were mixed to a final volume of 20 μL. The reaction was carried out for 30 min at 37 °C in a water bath. After the reaction was terminated, 4 μL of each of the 6× DNA Sampling Buffers were added to the gel. The pBR322 plasmid DNA alone was used as a negative control, and camptothecin was used as a positive control. Finally, the samples were loaded onto the gels and electrophoresed for about 1.5 h. After electrophoresis, the gel blocks were incubated in 1 mg/L ethidium bromide (EB) for 30 min and then visualized on the gel system.

#### 3.3.5. Cellular Thermal Shift Assay (CETSA)

For intact cell assays, Cal27 cells were treated with compound **7be** (8 μM) for 3 h before collection and washing several times with PBS buffer to avoid excess compound residues. Each cell suspension was dispensed into different 0.2 mL PCR tubes and then denatured for 3 min at different temperatures on a PCR instrument, and the cells were freeze-thawed twice with liquid nitrogen. After centrifugation, the supernatant was collected, mixed with an appropriate amount of up-sampling buffer and boiled at 95 °C for 6 min. The supernatant was detected by Western blot. For cell lysate experiments, Cal27 cells were collected with RIPA lysis buffer. Then compound **7be** (8 μM) was added to the supernatant and incubated at 25 °C for 2 h. After denaturation at different temperatures for 3 min, centrifugation was performed before collecting the supernatant, mixing with an appropriate amount of lysing buffer and boiling at 95 °C for 6 min, and then Western blot was performed to detect the supernatant.

#### 3.3.6. Cell Cycle Analysis

Cal27 cells were treated with compound **7be** for 24 h. Next, Cal27 cells were collected and suspended in ice ethanol at 4 °C for at least 24 h. Subsequently, Cal27 cells were incubated in RNase A and PI staining solution at 37 °C, protected from light, for 0.5 h. Finally, cells were detected by flow cytometry.

#### 3.3.7. Apoptosis Assay

Cal27 cells were treated with compound **7be** for 24 h. Tumor cells were then digested and collected. The resulting cells were incubated with Annexin V/PI light-avoidance staining for 0.5 h. Finally, different stages of apoptosis populations of Cal27 cells were immediately analyzed by flow cytometry.

#### 3.3.8. H&E Staining and Immunohistochemistry

Nude mouse organ and tumor tissues fixed with neutral formalin solution were paraffin-embedded according to the conventional processing method. Sections were cut to a thickness of approximately 5 μm, placed on glass slides and dewaxed in an oven at 60 °C, stained for H&E and immunohistochemistry and finally visualized with the Cytation 5 imaging system.

#### 3.3.9. Colony Formation Assay

The colony formation assay was used to evaluate the antiproliferative effect of compound **7be** on Cal27 cells. Cal27 cells were grown in 6-well plates at a density of 700 cells per well and then treated with different concentrations of compound **7be** for 48 h. The treatment medium was removed and fresh medium was added. After 2 weeks of culture, cells were fixed with 4% paraformaldehyde for 8 min and stained with crystal violet solution (10%). Finally, they were imaged with a camera.

#### 3.3.10. Wound Healing Assay

Cal27 cells in the logarithmic growth phase were inoculated into 6-well plates and cultured until the cells fused into a monolayer. A linear wound was then formed by scratching vertically with a 200 μL sterilized gun tip. The cells were washed 3 times with PBS to remove scraped off cells, and each well was incubated by adding medium. The initial scratched area was recorded with a microscope. Then different concentrations of the compounds to be tested were added, and the area of the scratches was recorded by microscopy 24 h later.

#### 3.3.11. Cellular Fluorescence Imaging

A certain amount of compound **7be** was dissolved with DMSO and configured into a 10 mmol/L masterbatch, and then compound **7be** (10 mM) was diluted with a PBS buffer solution (pH = 7.4) to 10 μM of test solution, ready for use. The test system was a PBS buffer solution (pH = 7.4) containing 5% DMSO.

A suspension of log-phase Cal27 cells was taken and inoculated into confocal dishes, 1 mL per well, with a drug control group (cells + drug) and an experimental group (cells + drug + dye). After the cells were adhered to the wall and treated with compound **7be** for 48 h, commercial mitochondrial Mito-Tracker Red dye and lysosomal Lyso-Tracker Red dye were added to each well for 5 min, and the supernatant was gently aspirated and discarded, and each well was washed three times with PBS, and shaking and mixing were performed so that the dye was washed out completely. The results were observed using a confocal microscope.

#### 3.3.12. RNA Extraction and Reverse Transcription Quantitative Real-time Polymerase Chain Reaction (RT-qPCR)

Cal27 cells were plated in a 6-well plate and incubated for 12 h at 37 °C in a humidified atmosphere with 5% CO_2_, then compound **7be** was added at final concentrations of 1 μM, 2 μM, 3 μM and 4 μM. After incubation for 24 h, cells were harvested, and the RNA was extracted according to the manufacturer’s instructions. Total RNA was used as a template for reverse transcription. The cDNA was obtained and applied directly for further qPCR. According to the manufacturer’s protocol, the real-time PCR was performed on a real-time PCR apparatus. The total volume of 20 μL of quantitative reaction mixtures contained 10 μL of SYBR Green qPCR Mix, 0.4 μL PCR Forward Primer (10 μM), 0.4 μL PCR Reverse Primer (10 μM) and 2 μL of cDNA (Appendix A). The *c-MYC* mRNA levels were normalized to the GAPDH mRNA level of each sample. Results of real-time PCR were analyzed using the 2^−ΔCt^ method. 

#### 3.3.13. In Vivo Antitumor Activity Assay

For this assay, 130 SPF-grade BALB/c mice (female, 4–6 weeks old, 18–20 g) were obtained from Chang Zhou Cavens Laboratory Animal Ltd. (Changzhou, China). The in vivo activity experiments were commissioned by Nanjing OG Pharmaceuticals, Co., Ltd. (Nanjing, China) All mice were housed in plastic cages, fed, and watered under standard conditions and air filtration (22 ± 2 °C; 12 h light/dark cycles). Under aseptic conditions, human oral cancer Cal27 cells were inoculated subcutaneously in the right axilla of BALB/c nude mice. When the tumors had grown to about 85 mm^3^, 24 hormonal mice with good growth status and good homogeneity of tumor size were selected and randomly divided into the following four groups of six mice each: (1) the model group; (2) the low-dose group; (3) the high-dose group; and (4) the positive drug group. The drug dose in the low-dose group was 10 mg/kg, which was injected intraperitoneally once every other day, for the duration of 21 days of administration; the drug dose in the high-dose group was 20 mg/kg, which was injected intraperitoneally once every other day, for the duration of 21 days of administration; and the drug dose in the cisplatin group was 2 mg/kg, which was injected intraperitoneally once every other day, for the duration of 21 days of administration. Normal tissues of the heart, liver, spleen, lung and kidney and half of the tumor tissues were fixed with 4% formalin solution and embedded in paraffin for histological analysis; the other half of the tumor tissues were preserved in liquid nitrogen, and the supernatant was extracted after milling the tissues and detected by Western blot.

#### 3.3.14. Statistical Analysis

All data were statistically analyzed using GraphPad Prism 8.0.2 software. Data were expressed as mean ± standard deviation (SD). One-way analysis of variance (ANOVA) and Bonferroni multiple comparisons were used. Results were considered statistically significant when the value was *p* < 0.05.

## 4. Conclusions

In summary, in order to develop new scaffolds of Topo I inhibitors with multiple mechanisms, 32 target derivatives were designed and synthesized. Their Topo I inhibitory activity and antiproliferative activity against five common cancer cell lines (NCI-H460, A549, ACHN, Cal27 and MGC-803) and one normal cell line (SV-HUC-1) were evaluated. The representative compound **7be**, with the most potent Topo I inhibitory activity and antiproliferative activity against the Cal27 (IC_50_ = 1.12 ± 0.22 μM), one of the oral cancer cell lines highly expressing Topo I, was selected for investigation of its anticancer mechanism. The results indicated that **7be** not only could bind to Topo I to exert its potent Topo I inhibitory activity, but also could repress the transcription of *c-MYC*, as well as inhibit the activation of PI3K/AKT/c-MYC and PI3K/AKT/NF-κB to arrest cells’ growth cycles, inhibiting cells’ colony formation and migration and inducing autophagy and apoptosis. In addition, in vivo studies demonstrated that **7be** exhibited an significant inhibitory effect on the growth of human oral cancer Cal27 xenograft tumors without significant toxic effects on major organs. Therefore, compound **7be** can be used as a potential leading compound for further development to a chemotherapeutic agent for oral cancer treatment.

## Data Availability

The original contributions presented in this study are included in the article/Appendix A. Further inquiries can be directed to the corresponding author.

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
