# Peer review of "Design, Synthesis and Bioactive Evaluation of Topo I/c-MYC Dual Inhibitors to Inhibit Oral Cancer via Regulating the PI3K/AKT/NF-κB Signaling Pathway"

_molecules, 2025, doi:10.3390/molecules30040894_

Round 1
Reviewer 1 Report
Comments and Suggestions for Authors
This article reports the design and synthesis of new molecules with a structural novelty of the sulfone type, as inhibitors of the Topo-I complex for the treatment of oral cancer.
The subject matter is interesting, and the article is well-written. The authors invested significant effort in conducting several experiments to corroborate their inhibition results. As a result of this effort, compound 7be is reported with an IC50 value of 1.12 µM in Cal27. Therefore, this compound could be considered a starting point for the development of subsequent series with improved inhibitory capacity (ideally in the nanomolar range) and a better selectivity profile for normal cells.
The work is easy to read, presenting several tables, graphs, and images that facilitate the presentation of results. The authors discuss a total of 61 bibliographic references, many of which are quite recent, which is appreciated.
From a chemical perspective, the primary novelty of the work is the introduction of the polycyclic system with the insertion of the sulfone group, which disrupts the absolute planarity of the system. However, this is debatable since the docking images show that the polycyclic system is, in fact, quite planar.
I believe this work contributes to the subject and to the journal, and I suggest accepting it provided the authors address the following comments:
-
The sulfone group is undoubtedly an innovation in this field. However, I wonder if the authors performed any quantum calculations on the polycycle structure to determine how much the new structures deviate from planarity. As mentioned above, they still appear to be completely coplanar structures. A simple minimization at the quantum level could show whether any changes occur. Alternatively, if the authors have a crystallographic structure of any derivative, it would be good to report it.
-
A comment on the comparative evaluation of fused benzothiophene systems instead of sulfone would be interesting. This could estimate how significant the presence of the two sp² oxygen atoms is in the polycyclic structure. While the docking shows potential hydrogen bonds to them, this remains a hypothesis.
-
I suggest that the authors define each acronym reported in Table 1 in its footnote. Although they are explained in the main manuscript, the current trend is for each table, graph, or figure to be self-explanatory so that the reader does not have to consult the main manuscript for the explanation of terms or abbreviations.
-
Figure 4 represents the authors’ effort to systematize a qualitative structure-activity relationship without resorting to QSAR methods. Here, a comment on the presence of the sulfone group is missed. Due to the absence of a comparative family without sulfone, definitive conclusions about the SOâ‚‚ group cannot be drawn. However, I would like to ask the authors if they have any comparative data from literature molecules using benzothiophene. If available, a comment on this in the manuscript would be interesting.
-
Regarding docking, I have several important observations:
- What is the protonation state of compound 7be in the simulation? Given the presence of a piperidine system, it should be monoprotonated with a net charge of +1, but this is not evident in the image where polar hydrogens are hidden. If the molecule was docked in a neutral state, the authors must redo the calculation.
- Was any comparative standard used for docking, such as Topotecan or a similar compound? Docking poses have little value when not compared with reported ligand crystals or at least with docking against a gold-standard compound. I urge the authors to clarify this and add the information to the supplementary material if no comparison has been made.
Author Response
Dear reviewer,
Thank you very much for your constructive comments and valuable suggestions. According to the comments, we have revised our manuscript carefully. Our responses to the comments were addressed point by point as follow and the corresponding changes were marked in red font in the revised manuscript.
- Here below is our description on revision according to you comments.
Reviewer #1: This article reports the design and synthesis of new molecules with a structural novelty of the sulfone type, as inhibitors of the Topo-I complex for the treatment of oral cancer.
The subject matter is interesting, and the article is well-written. The authors invested significant effort in conducting several experiments to corroborate their inhibition results. As a result of this effort, compound 7be is reported with an IC50 value of 1.12 µM in Cal27. Therefore, this compound could be considered a starting point for the development of subsequent series with improved inhibitory capacity (ideally in the nanomolar range) and a better selectivity profile for normal cells.
√ Response: Thanks for the reviewer’s positive comments.
The work is easy to read, presenting several tables, graphs, and images that facilitate the presentation of results. The authors discuss a total of 61 bibliographic references, many of which are quite recent, which is appreciated.
√ Response: Thanks for the reviewer’s positive comments.
From a chemical perspective, the primary novelty of the work is the introduction of the polycyclic system with the insertion of the sulfone group, which disrupts the absolute planarity of the system. However, this is debatable since the docking images show that the polycyclic system is, in fact, quite planar.
I believe this work contributes to the subject and to the journal, and I suggest accepting it provided the authors address the following comments:
- The sulfone group is undoubtedly an innovation in this field. However, I wonder if the authors performed any quantum calculations on the polycycle structure to determine how much the new structures deviate from planarity. As mentioned above, they still appear to be completely coplanar structures. A simple minimization at the quantum level could show whether any changes occur. Alternatively, if the authors have a crystallographic structure of any derivative, it would be good to report it.
√ Response: Thanks for the reviewer’s question. In our work the sulfone group was introduced in the polycyclic system to replace the carbonyl group which present in the famous Topo I inhibitor scaffold azaindenoisoquinolines reported by Cushman’s group. We mention in the manuscript that the insertion of the non-planar sulfone group might help to decrease the planarity of the polyaromatic ring skeleton. What we really mean was that compared to the carbonyl group, in which its oxygen and carbon atom are share the common plane with all the other atom at the polyaromatic ring skeleton, the two oxygens of the sulfone group is protruding above the polyaromatic ring skeleton, which make the ring is not a full plane so that not apt to intercalate into DNA to result in unexpected side effect. But not the mean that the present of the sulfone group would make the certain part of the polyaromatic ring deviate from planarity. To avoid of ambiguities, we have revised the expression in the manuscript.
- A comment on the comparative evaluation of fused benzothiophene systems instead of sulfone would be interesting. This could estimate how significant the presence of the two sp² oxygen atoms is in the polycyclic structure. While the docking shows potential hydrogen bonds to them, this remains a hypothesis.
√ Response: Thanks for the reviewer’s suggestion. At first we really tried to synthesize benzothiophene systems in a polycyclic skeleton, but found the products are very difficult to be dissolved in the common solvent that prevented the subsequent reaction. So the sulfone was induced. We also attempts to keep the oxidation product at the sulfoxide but was unsuccessful. The intermediate was very easy to be oxidated to sulfone.
- I suggest that the authors define each acronym reported in Table 1 in its footnote. Although they are explained in the main manuscript, the current trend is for each table, graph, or figure to be self-explanatory so that the reader does not have to consult the main manuscript for the explanation of terms or abbreviations.
√ Response: Thanks for the reviewer’s suggestion. We have added all the abbreviations reported in Table 1 in its footnote.
- Figure 4 represents the authors’ effort to systematize a qualitative structure-activity relationship without resorting to QSAR methods. Here, a comment on the presence of the sulfone group is missed. Due to the absence of a comparative family without sulfone, definitive conclusions about the SOâ‚‚ group cannot be drawn. However, I would like to ask the authors if they have any comparative data from literature molecules using benzothiophene. If available, a comment on this in the manuscript would be interesting.
√ Response: Thanks for the reviewer’s suggestion. Some synthetic benzothiophene s ystems with structure similar to intermediate 5 have been reported by Kalugin group (Synthesis of substituted pyrido[3′,2′:4,5]thieno[3,2c]isoquinolin5(6H)ones and their sulfinyl and sulfonyl derivatives, Russ. Chem. Bull., Int. Ed., 2017, 66, 523-530; The synthesis of substituted 5-aminopyrido[3´,2´:4,5]thieno[3,2-c]isoquinolines and their sulfi nyl and sulfonyl derivatives, Russ. Chem. Bull., Int. Ed., 2018, 67, 902-911; Synthesis of 5-aminopyrido[3´,2´:4,5]thieno[3,2-c]isoquinoline derivatives from 3-cyanopyridine-2(1H)-thiones and 2-(chloromethyl)benzamide, Russ. Chem. Bull., Int. Ed., 2018, 67, 1492-1499), but there are no any data about their bioactivity. So we do not have any comparative QSAR data from molecules using benzothiophene.
- Regarding docking, I have several important observations:
What is the protonation state of compound 7be in the simulation? Given the presence of a piperidine system, it should be monoprotonated with a net charge of +1, but this is not evident in the image where polar hydrogens are hidden. If the molecule was docked in a neutral state, the authors must redo the calculation.
Was any comparative standard used for docking, such as Topotecan or a similar compound? Docking poses have little value when not compared with reported ligand crystals or at least with docking against a gold-standard compound. I urge the authors to clarify this and add the information to the supplementary material if no comparison has been made.
√ Response: Thanks for the reviewer’s suggestion. In the physiological conditions the piperidine system might probably be monoprotonated with a net charge of +1. Since the docking software can automatically conduct energy optimization of the molecule based on the physiological conditions, so in the docking study, the neutral state of the compound 7be was used to energy optimization and docking.
To give a comparative standard for docking, the molecular docking results of indenoisoquinoline, a Topo I inhibitor scaffold reported by Cushman’s group with structurally similar to the compound 7be have been added in the manuscript, and the results indicated that it really share same binding mode with the Topo I/DNA and gave the same docking result reported in the literature when used a different docking software (Staker, B. L.; Feese, M. D.; Cushman, M.; Pommier, Y.; Zembower, D.; Stewart, L.; Burgin, A. B. Structures of Three Classes of Anticancer Agents Bound to the Human Topoisomerase I-DNA Covalent Complex, J. Med. Chem. 2005, 48, 2336-2345). And some corresponding description have been added into the manuscript.
Reviewer 2 Report
Comments and Suggestions for Authors
The authors in their manuscript reported the synthesis of a large group of (32) pyridothieno[3,2-c]isoquinoline 11,11-dioxide. The substrate for the synthesis of these compounds was obtained in a seven-step process starting from chlorocyanopyridine. The authors performed a very large work on the synthesis of such a large group of new substances, for which the molecular target was DNA 13 topoisomerase I (Topo I). The authors confirmed the preparation of the assumed group of compounds by NMR analyses (1H and 13C), HRMS and X-ray structural analysis of selected derivatives. This is confirmed by the analyses included in the supplement. The only drawback of these syntheses is the yield, in the case of most compounds it is a few to a dozen percent, the authors did not attempt to optimize the reaction conditions, or did not write about it.
In my opinion the manuscript is interesting and valuable, all parts (introduction, parts about the experiment, discussion of results and conclusions) are properly presented. The authors received 32 new substances, some of which showed properties comparable or better than the reference drugs used in the study. The manuscript is careful in terms of editing. A supplement with the results of biological and spectroscopic tests is included. After introducing corrections (below) it can be published in Molecules.
1. Did the authors synthesize substances 1-6 according to literature procedures or did they develop their own syntheses? In the case of these compounds, it would be necessary to refer to the literature because these are known substances. Melting points and literature references for these substances should be added in the experimental part.
2. Melting points notation e.g. 268~270 °C, should be changed by the authors to 268-270 °C. Similarly 7aa ~ 7al, 7ba ~ 7bj, 7ca ~ 7cj should be changed to 7aa - 7al, 7ba - 7bj, 7ca - 7cj (Scheme 1). Authors should also remove duplicate numbering from references.
3. In chemical names such as: Pyrido[3',2':4,5]thieno[3,2-c]isoquinolin-5(6H)-one 11,11-dioxide H should be written in italics (Pyrido[3',2':4,5]thieno[3,2-c]isoquinolin-5(6H)-one 11,11-dioxide).
Author Response
Dear reviewer,
Thank you very much for your constructive comments and valuable suggestions. According to the comments, we have revised our manuscript carefully. Our responses to the comments were addressed point by point as follow and the corresponding changes were marked in red font in the revised manuscript.
Here below is our description on revision according to the your comments.
Reviewer #2: The authors in their manuscript reported the synthesis of a large group of (32) pyridothieno[3,2-c]isoquinoline 11,11-dioxide. The substrate for the synthesis of these compounds was obtained in a seven-step process starting from chlorocyanopyridine. The authors performed a very large work on the synthesis of such a large group of new substances, for which the molecular target was DNA 13 topoisomerase I (Topo I). The authors confirmed the preparation of the assumed group of compounds by NMR analyses (1H and 13C), HRMS and X-ray structural analysis of selected derivatives. This is confirmed by the analyses included in the supplement. The only drawback of these syntheses is the yield, in the case of most compounds it is a few to a dozen percent, the authors did not attempt to optimize the reaction conditions, or did not write about it.
√ Response: Thanks for the reviewer’s reminding. In fact, different solvents, including DMF, DMSO and toluene have been tried to optimize the reaction, but the yield of some products were very difficult to be improved. We found that the intermediate 6 is very sensitive to water. We speculate that the substrate with unsatisfactory might due to the different water content in the material amine.
In my opinion the manuscript is interesting and valuable, all parts (introduction, parts about the experiment, discussion of results and conclusions) are properly presented. The authors received 32 new substances, some of which showed properties comparable or better than the reference drugs used in the study. The manuscript is careful in terms of editing. A supplement with the results of biological and spectroscopic tests is included. After introducing corrections (below) it can be published in Molecules.
√ Response: Thanks for the reviewer’s positive comments.
- Did the authors synthesize substances 1-6 according to literature procedures or did
they develop their own syntheses? In the case of these compounds, it would be necessary to refer to the literature because these are known substances. Melting points and literature references for these substances should be added in the experimental part.
√ Response: Thanks for the reviewer’s suggestion. In our synthesis scheme the intermediate 3 and 5 was synthesized according to literature procedures with slight optimization and the literature references have been added in the manuscript. Procedures for synthesis of the other substances was developed by ourself. The melting points of the known or unknown substances also was measured and added in the experimental part.
- Melting points notation e.g. 268~270 °C, should be changed by the authors to 268-270 °C. Similarly 7aa ~ 7al, 7ba ~ 7bj, 7ca ~ 7cj should be changed to 7aa - 7al, 7ba - 7bj, 7ca - 7cj (Scheme 1). Authors should also remove duplicate numbering from references.
√ Response: Thanks for the reviewer’s suggestion. We have changed the symbol "~" to "-" and removed the duplicate numbering in the references.
- In chemical names such as: Pyrido[3',2':4,5]thieno[3,2-c]isoquinolin-5(6H)-one 11,11-dioxide H should be written in italics (Pyrido[3',2':4,5]thieno[3,2-c]isoquinolin-5(6H)-one 11,11-dioxide).
√ Response: Thanks for the reviewer’s reminding. We have changed the letter "H" to italics.